# Experience modulates gaze behavior and the effectiveness of information pickup to overcome the inversion effect in biological motion perception

Xiaoye Michael Wang[1]*, Zhichen Feng[2], Mingming Yang[2], Jing Samantha Pan[3,4], Margaret A. Wilson[5], Qin Zhu[2]

1 Faculty of Kinesiology and Physical Education, University of Toronto, Toronto, ON, Canada, 2 Division of Kinesiology and Health, University of Wyoming, Laramie, WY, United States of America, 3 Department of Psychology, Sun Yat-sen University, Guangzhou, China, 4 Guangdong Provincial Key Laboratory of Social Cognitive Neuroscience and Mental Health, Guangzhou, China, 5 Department of Theatre and Dance, University of Wyoming, Laramie, WY, United States of America

* michaelwxy.wang@utoronto.ca

**Data Availability Statement:** Data for this study can be accessed via https://osf.io/sv3qc/.

## Abstract

The inversion effect in biological motion suggests that presenting a point-light display (PLD) in an inverted orientation impairs the observer's ability to perceive the movement, likely due to the observer's unfamiliarity with the dynamic characteristics of inverted motion. Vertical dancers (VDs), accustomed to performing and perceiving others to perform dance movements in an inverted orientation while being suspended in the air, offer a unique perspective on this phenomenon. A previous study showed that VDs were more sensitive to the artificial inversion of PLDs depicting dance movements when compared to typical and non-dancers if given sufficient dynamic information. The current study compared the gaze behaviors of non-dancers, typical dancers, and VDs when observing PLDs of upright and inverted dance movements (either on the ground or in the air) to determine if the PLDs were artificially inverted. Behavioral results replicated the previous study, showing that VDs were more sensitive in detecting inverted movements. Eye-tracking data revealed that VDs had longer fixations, primarily directed at the depicted dancer's pelvic area. When performing movements in the air, the depicted dancer was suspended via a harness around their pelvis, providing unique dynamic information that specified the movement's canonical orientation. In contrast, although typical dancers also attended to the pelvic area, their lack of experience with perceiving and performing vertical dance movements limited their ability to interpret the dynamic information effectively. These findings highlight the role of specialized visuomotor experience in enhancing biological motion perception and have implications for training techniques that leverage visual strategies to improve performance in complex or unfamiliar movement contexts.

**Funding:** The author(s) received no specific funding for this work.

**Competing interests:** The authors have declared that no competing interests exist.

## Introduction

As terrestrial animals, gravity has been an indispensable element in hominid's evolutionary past, present, and future [1]. Its significance not only lies in a fallen apple mentioned in a textbook, but it also manifests in learning and controlling human movements. Any human movement on Earth is subject to the influence of the entire force environment, the presence of gravity is a key aspect in determining how a person moves. The influence of gravity (or a lack thereof) can be readily seen in how differently the astronauts move in space, where performing motor tasks in a microgravity environment without having their feet on the ground could be challenging. Past research has shown that human observers could readily take advantage of the dynamic (force) information embedded in human movements to perform various types of perceptual judgment, such as the actor's gender [2,3] and identity [4–6], as well as the effect of actions [7]. To control for other visual features (e.g., configuration information such as shape and size) that convey information about the actor and the action, point-light displays (PLD) are used to isolate the dynamic and configural information of the movement [8,9]. Point-light displays represent a person or an animal using a series of moving points placed at different major joints (e.g., shoulders, elbows, wrists, pelvis, knees, and ankles). According to the *kinematic specification of dynamics* theory [10], how an object moves (kinematics) is determined by the object's entire force environment (dynamics), including gravity. Therefore, altering the force environment would in turn perturb how an object moves, as in the case of an astronaut in space.

Reducing gravitational force is not the only method to perturb the force environment in which a person moves. Studies have shown that simply inverting the PLD could impair the observer's ability to perceive biological motion, termed the *inversion effect* of biological motion [11–13]. For instance, Shipley [12] used PLDs to present a person walking where the display was either in the movement's canonical orientation or upside down. In this context, the artificial inversion also inverted the direction of gravity in the display, altering the embedded dynamics of the movement. Results showed that the artificial inversion reduced the participants' ability to identify meaningful movement patterns in the display, suggesting the importance of dynamic information in specifying natural movements.

While the inversion effect yields impairment to biological motion perception, a recent study suggested that visuomotor experience with inverted movements could help the observers to overcome this adverse effect [14]. This study focused on the unique visuomotor experience of vertical dancers (VD), as compared to typical (TD) and non-dancers (ND). Vertical dance is a form of aerial modern dance in which the dancers are suspended in mid-air wearing a harness around the waist and hips and perform various movements either in an upright or upside-down orientation. As a result, VDs have both extended visual experience of observing others performing movements in an inverted orientation and the motor and proprioceptive experience of performing such movements themselves. In this study, a series of dance movements were recorded either on the ground or in the air and presented as PLDs that were either in the movement's canonical orientation or artificially inverted. Then, a mixture of original and inverted PLDs was presented either in their entirety (i.e., with sufficient dynamic information) or as a one-second clip that only depicts the final form of the movement (i.e., with limited dynamic information) for observers to detect the artificial inversion. Results showed that, while TD and ND had difficulties discerning the artificial inversion of the movements performed in the air, VD could readily do so when given sufficient dynamic information.

Unlike walking, dance movements are more complex, involving the coordination between different body segments and using both upper and lower limbs to demonstrate diverse postures. With their feet off the ground and body suspended in the air, vertical dancers have more

degrees of freedom to use their trunks and limbs for creative performance. Therefore, the unexpectedly greater range of motion associated with the trunk and limbs in vertical dance, particularly in the inverted orientation, could have confused observers who typically see movements performed on the ground, limiting their ability to judge and interpret the vertical dance movements. Although Wang and colleagues [14] demonstrated that VDs were able to use dynamic information in detecting the inversion effect, the source of such dynamic information based on which the judgment was made remains unclear. Identifying the source of dynamic information in vertical dance could not only advance the scientific understanding of human movement perception but also yield insights into the ways through which dancers can improve the expressiveness and impact of their performances by focusing on their own or other's body segments that convey movement dynamics (e.g. force, intention, or emotion).

One way to identify the source of dynamic information in vertical dance is by studying the observer's gaze behaviors in relation to the PLDs. Gaze behaviors are dictated by five factors, including bottom-up salience, top-down feature guidance, scene structure and meaning, the previous history of search over timescales, and the relative value of the targets and distractors [15]. For instance, radiologists rely on years of education and training to adopt a global-focal search strategy, allowing them to quickly detect salient abnormalities in medical images by matching these features to structures stored in their memory [16]. Similarly, due to their extensive experience performing and observing dance movements in both upright and inverted orientations, VDs are potentially more attuned to the dynamic information in the display, which helps them overcome the inversion effect.

Eye-tracking technology is commonly employed to study gaze behaviors, allowing researchers to differentiate between the gaze patterns of experts and novices and identify the visual information picked up during the process. Stevens et al. [17] examined the fixations and saccades of expert and novice dancers when viewing a dance film. The authors found that, compared to novices, experts demonstrated shorter fixation durations and faster saccades, indicative of a more deliberate and efficient search strategy. Moreover, the gaze locations differed: experts attended mostly to the head of the dancers, while novices attended equally to the head, neck, torso, and arms. These results were replicated by Ponmanadiyil and Woolhouse [18], who compared the gaze behavior between experienced and inexperienced dancers when observing Indian Bharatanatyam dance. Given that VDs are more capable than non-VDs at detecting inverted dance movements, it raises the question of whether dancers with and without VD experience employed distinct gaze behaviors when tasked with detecting artificially inverted movements.

The current study aimed to extend the findings from Wang et al. [14] by incorporating eye-tracking measures to identify the sources of dynamic information on which VDs rely when detecting inverted movements. It was hypothesized that, compared to TDs and NDs, VDs would engage in more frequent fixations and saccades, reflective of a more efficient process to identify the source of relevant dynamic information embedded in the PLDs for the perceptual judgment task. Moreover, a spatial analysis of fixation points relative to the inherently information-rich area of the PLDs was expected to reveal the specific sources of dynamic information that enable VDs to make accurate judgments.

## Methods

### Participants

37 adults participated in this experiment, including 12 non-dancers (ND; age M = 23.00 years, SD = 2.22), 13 typical dancers (TD; age M = 22.17 years, SD = 1.47), and 12 vertical dancers (VD; age M = 32.90 years, SD = 18.56). Typical dancers had an average of 12.25 years of dance

experience, whereas VDs had an average of 9.82 years of vertical dance experience. The experiment was in accordance with the Declaration of Helsinki and was approved by the University of Wyoming Institutional Review Board (IRB). All participants provided written informed consent prior to the experiment.

## Stimuli and apparatus

The visual stimuli used in this study were identical to that of Wang et al. [14]. A male and a female professional vertical dancer performed a series of dance movements. Each dancer was fitted with 13 optical markers. The markers' 3D positions were captured using a Vicon Motion Capture system with eight Bonita Optical cameras at a 160 Hz sampling frequency. The markers correspond to 13 major joints, including the head, the left and right shoulders, elbows, wrists, hip joints, knees, and ankles. The 3D coordinates of each movement were projected onto a 2D plane to generate the animation. Both dancers provided their informed consent for the motion capture and the subsequent use of the motion capture data for the experiment.

Each dancer performed 20 movements, where 10 movements were performed on the ground and the other 10 were performed in the air while being suspended with a harness (Table 1). Fig 1 demonstrates the basic distinctions between different types of movements. The air and ground movements were selected such that each movement pair is similar, if not identical, in form (i.e., same configural information), but only differ in terms of their recording conditions (i.e., on the ground or in the air). Moreover, of the ten ground-air pairs, five pairs' canonical orientations were identical (congruent, i.e., both the ground- and air-based movements were performed in an upright/inverted orientation), whereas the other five pairs had opposite canonical orientations (incongruent, i.e., the ground movement was performed in an upright/inverted orientation whereas the air movement was performed in an inverted/upright orientation). Movements performed by one dancer were kept in their original sequence (mean duration = 4.05 s, SD = 1.10, range = [1.25, 5.94] s). Presented in their entirety, these movements provided complete dynamic information. In contrast, movements performed by the other dancer were trimmed such that only the final form of the movement was available for one second, where the dancer remained relatively static. These movements only provided limited dynamic information and required the participants to rely more on the configuration information from the PLDs to perform the judgment.

**Table 1. List of dance movements.**

| Ground | Air | Congruency |
|---|---|---|
| Arabesque Fondu | Inverted Arabesque Fondu | Incongruent |
| Attitude devant over forced arch/arms forward | Inverted attitude devant/arms forward | Incongruent |
| Développé à la seconde | Inverted développé à la seconde | Incongruent |
| Italian Pas de Chat | Inverted Pas de Chat | Incongruent |
| X–Plié/extend | Inverted X–Plié/extend | Incongruent |
| Attitude spiral | Side-lying attitude | Congruent |
| Cambre Back B+ | High release one leg attitude | Congruent |
| Plank to V-sit | Plank to V-sit | Congruent |
| Supported handstand flex/point | Inversion hook foot | Congruent |
| Side Plank–hip lower and lift | Side-lying limbs lower/lift | Congruent |

A list of dance movements performed on the ground or in the air with the congruency of their canonical orientations.

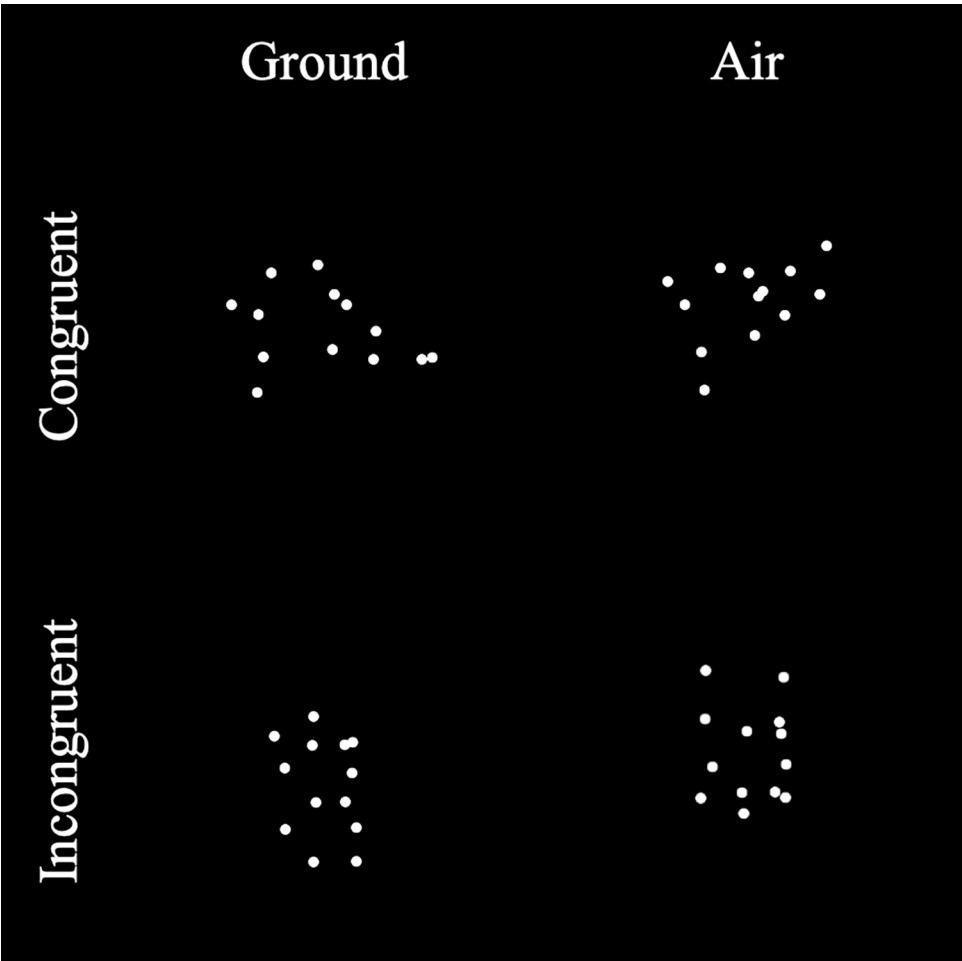

**Fig 1.** Illustration of the ground and air movements when the movement pair was incongruent (top; ground: Side Plank–hip lower and lift; air: Side lying limbs lower/lift) or congruent (bottom; ground: X–Plié/extend; air: Inverted X–Plié/extend). More details can be found in a video demo (https://youtu.be/6vLQgQpy-uc) from the previous study [14].

The experimental stimuli were presented on a Dell 22-inch monitor (E2222H) with a 1680 × 1050 resolution. The monitor was placed 57 cm in front of the participant while the participant rested their head on a chinrest. The experiment was coded in Python using functionalities from PsychoPy [19]. Eye tracking was performed using Tobii Pro Fusion, which was attached to the bottom of the monitor. Raw eye tracking data, measured based on the screen's pixel coordinate system ($x$-axis: 0–1680 pixels; $y$-axis: 0–1050 pixels), were collected using PyGaze, a Python-based open-source toolbox that offers an interface with Tobii devices [20].

## Procedure

After providing their informed consent, the participants were first introduced to the task using a laptop computer. They were presented with a pair of ground-air movements that were not used in the actual experiment. The movement animations represented each joint as a 10-pixel white circle displayed against a black background (Fig 1). The tutorial used these movements to illustrate the difference between ground and air movements and between the different canonical orientations (upright or inverted). The participants were told that the displayed

movements could be artificially inverted from their canonical orientations, and their task was to identify whether the artificial inversion had occurred in an animation. They responded by pressing the F key for inverted and the J key for not inverted. Then, the participants did four practice trials using the demonstration movements and received feedback.

After the tutorial, the participants were seated in front of the monitor and placed their chin on a custom-made chinrest. The experiment started with calibrating the eye tracker, which was performed using PyGaze's calibration method. The subsequent perceptual judgment task was blocked by Dynamic Type (complete or limited), presented in a random order. In each block, there were 20 movements (5 movements × 2 Recording Types [ground or air] × 2 Congruency [congruent or incongruent]) that were either presented as-is or artificially inverted, resulting in a total of 40 trials, presented in random order. In each trial, each movement animation was repeated three times, after which the participants were asked to determine whether the animation was artificially inverted. Combining two blocks, the entire experiment contained 80 trials and took approximately 30 minutes to complete.

### Data analysis

**Perceptual judgment.**   Perceptual judgment was analyzed using signal detection theory [21] to derive the participants' sensitivity to artificial inversion. For each participant, trials were grouped based on Dynamic Types (complete or limited), Congruency (congruent or incongruent), and Recording Types (ground or air). Each group contained 10 trials with responses of yes for the displayed movement that was artificially inverted from their canonical orientation and no for the movement that was not inverted. The artificial inversion was considered as the signal and therefore, a hit represents a yes response when the movement was artificially inverted, whereas a false alarm corresponds to a yes response when the movement was not inverted. The sensitivity, d', was calculated as the difference in z-scores between hit and false alarm [22]. A d' value greater than 0 indicates the participants were able to identify the artificial inversion above the chance level, whereas a negative d' value suggests that the participants considered the movements displayed in their canonical orientation as being artificially inverted. Subsequently, a mixed-design analysis of variance was used to analyze the effects of one between-subject factor of Experience (three levels: ND, TD, and VD) and three within-subject factors of Dynamic Type (two levels: complete and limited), Congruency (two levels: congruent and incongruent), and Recording Type (two levels: ground and air) on the derived d' values. Greenhouse-Geisser corrections were applied to effects that violate the sphericity assumption and were indicated by the decimal degrees of freedom values.

**Eye tracking.**   Raw eye-tracking data were processed using PyTrack, a Python-based eye-tracking analysis toolkit [23]. Fixation threshold was set at 2° with a minimum duration of 100 ms and saccades were identified based on a velocity threshold of 60°/s with a minimum duration of 30 ms. Fixation frequency and duration as well as saccade frequency were derived for each trial. To further examine the relationship between fixation points and the PLD, an additional spatial analysis of the gaze patterns was performed. The pelvic girdle including the hip joints plays an important role in the functional movement of dancers because it connects the upper and lower body and contributes to posture and balance control [24]. When performing dance movements midair while suspended in a harness, movement of the pelvis is restricted by the rope connecting the harness to the ceiling. In contrast, the pelvis is less constrained when performing the same dance movements on the ground. Therefore, the pelvic girdle may serve as an inherently information-rich area for the observer (especially dancers) to differentiate between air and ground movements. To examine this hypothesis, the pixel coordinates corresponding to the location of the fixation and the location of the pelvic center were recorded.

The pelvic center was approximated as the midpoint between the two markers at the hip joints. To derive a spatial measure for the relationship between fixation and the PLD, the Euclidean distance (in pixels) between the pelvic center and the fixation point was calculated for each animation frame during the fixation period. Then, the mean distances, referred to as the mean Euclidean distance, across all fixation periods were calculated for every trial.

Based on findings of a previous study [14], it was expected that, due to their extended visuomotor experience, VDs were more sensitive to the inversion of air movements than NDs and TDs when given complete dynamic information and regardless of the movement's congruency. In this context, the goal of eye tracking analysis is to investigate the source of the dynamic information in the animation that VDs focused on to achieve higher sensitivity. Therefore, statistical analysis of eye tracking data, including gaze events and the mean Euclidean distance, will only use data from the complete dynamic condition and will not include Congruency as a separate factor. Therefore, a series of mixed-design ANOVA with one between-subject factor (Experience) and one within-subject factor (Recording Type) was performed on gaze events (i.e., saccade frequency, fixation frequency, and fixation duration) and the Euclidean fixation distance.

## Results and discussion

### Perceptual judgment

Table 2 shows the mixed-design ANOVA results for d'. Because the current study involved direct replication of a previous study [14], discussions of perceptual judgment results will primarily focus on significant effects related to Experience. For the significant main effect of Experience, VD (mean = 0.84, SE = 0.095) had significantly greater d' than ND (mean = 0.36, SE = 0.095, $p < 0.01$), while VD's difference with TD (mean = 0.56, SE = 0.092) failed to reach the traditional level of significance ($p = 0.097$). There was no difference between TD and ND ($p = 0.30$). The current finding is consistent with the general patterns of results reported earlier, where due to their extended visuomotor experience with inverted movements, VDs demonstrated an advantage over TDs and NDs with regard to differentiating between physically and artificially inverted movements.

For the significant main effect of Recording Type, sensitivity to the artificial inversion of ground movement (mean = 1.27, SE = 0.10) was significantly greater than that of air movement (mean = -0.10, SE = 0.11, $p < 0.001$). As expected, participants across all experience levels were more familiar with movements performed on the ground and, therefore, were more sensitive to the artificial inversion of ground movements. Interestingly, the significant two-way interaction between Experience and Recording Type and the three-way interaction between Experience, Recording Type, and Congruency revealed the impact of visuomotor experience on perceptual judgment. Because the significant three-way interaction encompasses the two-way interaction, only the post-hoc analysis of the three-way interaction will be discussed (Fig 2). Experience was used as a simple effect to reduce the total number of pairwise comparisons. For the congruent movements, where the air and ground movement counterparts were performed in the same canonical direction, there was no difference between any Experience pairs when the movement was performed on the ground (VD–TD: $p = 0.98$; VD–ND: $p = 0.91$; TD–ND: $p = 0.97$). For the congruent air movements, VD (mean = 0.63, SE = 0.16) showed greater sensitivity than TD (mean = 0.04, SE = 0.16, $p < 0.05$) and ND (mean = 0.08, SE = 0.16, $p = 0.06$), albeit the latter only approached the traditional level of significance. For the incongruent air movements, VD (mean = 0.71, SE = 0.36) had significantly greater sensitivity than TD (mean = -0.73, SE = 0.35, $p < 0.05$) and ND (mean = -1.34, SE = 0.36, $p < 0.01$). The noticeable negative d' values for TD and ND are indicative that both

**Table 2. Results of the mixed-design ANOVA on d'.**

| Effects | DF1 | DF2 | $F$ | $\eta_p^2$ | $p$ | |
|---|---|---|---|---|---|---|
| Experience | 2 | 34 | 6.47 | 0.28 | < 0.01 | ** |
| Recording type | 1 | 34 | 59.17 | 0.64 | < 0.001 | *** |
| Experience × Recording type | 2 | 34 | 8.29 | 0.33 | < 0.01 | ** |
| Congruency | 1 | 34 | 0.09 | 0.00 | 0.77 | |
| Experience × Congruency | 2 | 34 | 1.06 | 0.06 | 0.36 | |
| Dynamic type | 1 | 34 | 40.12 | 0.54 | < 0.001 | *** |
| Experience × Dynamic type | 2 | 34 | 6.69 | 0.28 | < 0.01 | ** |
| Recording type × Congruency | 1 | 34 | 18.39 | 0.35 | < 0.001 | *** |
| Experience × Recording type × Congruency | 2 | 34 | 5.33 | 0.24 | < 0.01 | ** |
| Recording type × Dynamic type | 1 | 34 | 1.38 | 0.04 | 0.25 | |
| Experience × Recording type × Dynamic type | 2 | 34 | 0.82 | 0.05 | 0.45 | |
| Congruency × Dynamic type | 1 | 34 | 10.35 | 0.23 | < 0.01 | ** |
| Experience × Congruency × Dynamic type | 2 | 34 | 0.53 | 0.03 | 0.59 | |
| Recording type × Congruency × Dynamic type | 1 | 34 | 4.81 | 0.12 | < 0.05 | * |
| Experience × Recording type × Congruency × Dynamic type | 2 | 34 | 1.01 | 0.06 | 0.37 | |

\* $p < 0.05$

\*\* $p < 0.01$

\*\*\* $p < 0.001$.

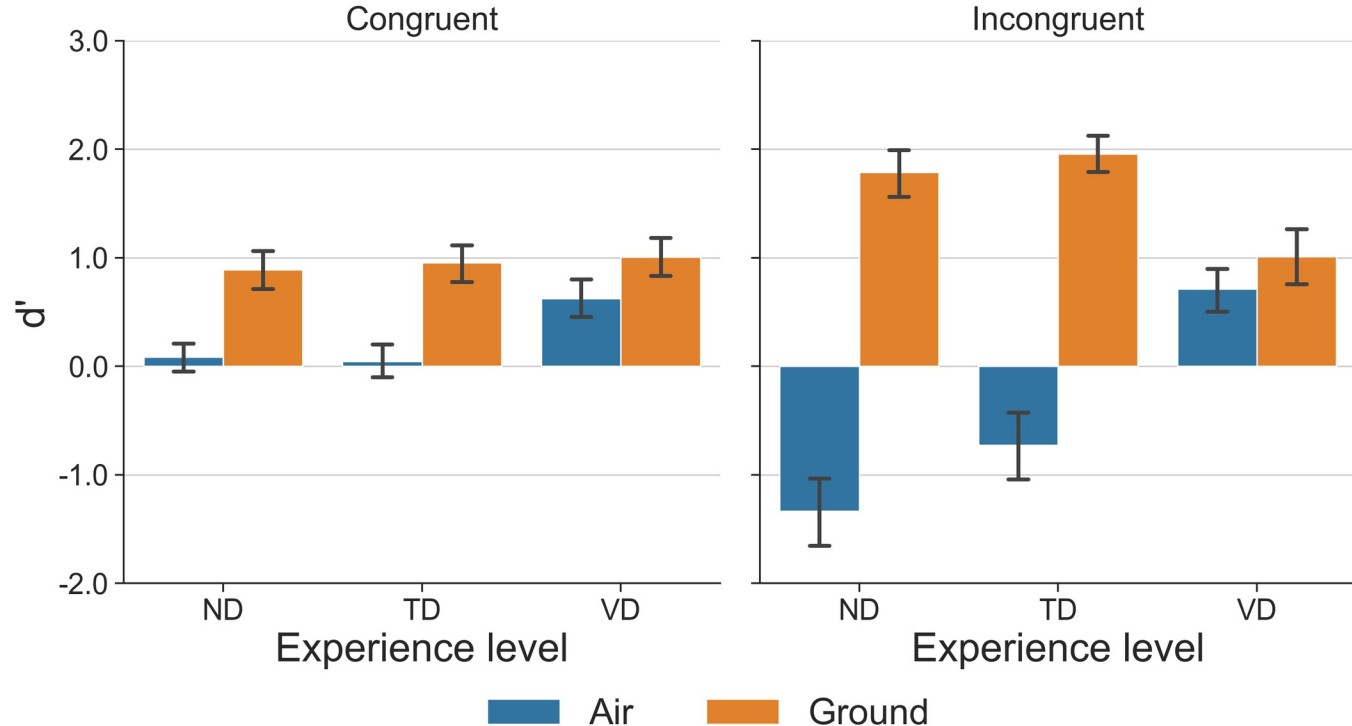

**Fig 2.** Mean d' as a function of Experience (*x*-axis: ND: Non-dancers, TD: Typical dancers, VD: Vertical dancers) for different Recording Types (hue) for the congruent (left) and incongruent (right) air-ground movement pairs. Error bars represent the standard error of the means.

groups mistook un-inverted air movements (in an upside-down canonical orientation) as being artificially inverted. Because the canonical orientation of the incongruent air movements was upside down, the large negative d' values suggest that TD and ND simply considered the upright orientation as the movement's canonical orientation and judged every display with an upside-down dancer as being artificially inverted, regardless of the actual orientation in which movement was performed. This reflected the unfamiliarity of TD and ND with vertical dance and a bias towards movement performed in an upright orientation. For the incongruent ground movements, VD (mean = 1.01, SE = 0.22) had significantly lower sensitivity than TD (mean = 1.96, SE = 0.21, $p < 0.05$) and ND (mean = 1.79, SE = 0.22, $p < 0.05$). This could also reflect the stronger bias of TD and ND where they generally considered inverted movements as being artificially inverted from the ground movements and the lack of a similar bias of VD.

For the main effect of Dynamic Type, sensitivity in the Complete condition (mean = 0.87, SE = 0.08) was significantly higher than that in the Limited condition (mean = 0.30, SE = 0.05, $p < 0.001$). Importantly, the significant interaction between Experience and Dynamic Type showed how well different participant groups utilized the dynamic information. For the Limited condition, no difference was observed between any Experience pair (VD–TD: $p = 0.71$; VD–ND: $p = 0.82$; TD–ND: $p = 0.98$). However, for the Complete condition, VD (mean = 1.32, SE = 0.15) had greater sensitivity than TD (mean = 0.85, SE = 0.14, $p = 0.073$) and ND (mean = 0.43, SE = 0.15, $p < 0.001$), albeit the former only approached the traditional level of significance. This finding suggests that, compared to TDs and NDs without any vertical dance experience, VDs were more capable of utilizing the dynamic information in the display to perform correct judgment, which replicated previous results [14].

## Eye tracking analysis

The primary goal of this study was to examine gaze patterns to figure out how VDs utilized dynamic information to distinguish artificial inversions in air and ground movements. To this end, only data from the Complete condition were used. A series of mixed-design ANOVAs with Experience (between-subject) and Recording Type (within-subject) as factors were performed for the number of saccades, number of fixations, and fixation duration. Congruency was not included in the analysis because it is more relevant to the intrinsic bias of each participant group, instead of their ability to use the dynamic information in the display.

For the number of saccades, there was only a significant main effect of Recording Type, $F(1, 34) = 20.89, p < 0.001, \eta_p^2 = 0.38$, while neither Experience, $F(2, 34) = 1.22, p = 0.31$, $\eta_p^2 = 0.07$, nor the interaction, $F(2, 34) = 0.02, p = 0.98, \eta_p^2 = 0.001$, were significant. There were significantly more saccades with the air movement (mean = 16.9, SE = 0.95) than with the ground movement (mean = 15.2, SE = 0.78, $p < 0.001$). This finding suggests an increased level of visual exploration in the air condition, which also corresponded to an increased task difficulty. In other words, because of the air movements' uniqueness, the participants across all experience levels engaged in more frequent gaze shifts to identify relevant visual information to perform the judgment. For the number of fixations, there was again only a significant main effect of Recording Type, $F(1, 34) = 34.62, p < 0.001, \eta_p^2 = 0.50$, but not Experience, $F(2, 34) = 0.01, p = 0.99, \eta_p^2 < 0.001$, or the interaction, $F(2, 34) = 0.34, p = 0.72$, $\eta_p^2 = 0.02$. There were more fixations with the air movements (mean = 15.4, SE = 0.68) than with the ground movement (mean = 13.9, SE = 0.58, $p < 0.001$). This finding is consistent with the increased saccade frequency with the air movement, further demonstrating more visual exploration.

Finally, for the fixation duration, Recording Type was significant, $F(1, 34) = 5.00, p < 0.05, \eta_p^2 = 0.13$. Although the main effect of Experience was not significant, $F(2, 34) = 0.74$, $p = 0.48, \eta_p^2 = 0.04$, the interaction between Experience and Recording Type was, $F(2, 34) = 3.68, p < 0.05, \eta_p^2 = 0.18$. As with other gaze metrics, fixation duration for air movements (mean = 401 ms, SE = 9.60) was significantly greater than that for ground movements (mean = 394 ms, SE = 9.27, $p < 0.05$). More crucially, for the significant interaction, there was no difference between the air and ground movements for TDs ($p = 0.51$) or NDs ($p = 0.19$). However, for VDs, the fixation duration for the air movements (mean = 403 ms, SE = 16.84) was significantly longer than that for the ground movements (mean = 384 ms, SE = 16.26, $p < 0.01$). The longer fixation duration for VDs with the air movement indicates that VDs concentrated on certain features of the air movements more than in the ground movements. In conjunction with their general high sensitivity to the artificial inversion in the air movement and a lack of difference in their saccade and fixation frequencies, this finding implies that VDs were able to identify the potential area of interest more readily in the display and extrapolate the relevant information that revealed the canonical orientation of the movement. Subsequently, the spatial patterns of the fixation were analyzed to explore the sources of visual information that VDs relied on to perform the judgment.

## Spatial analysis

The spatial analysis of gaze patterns focused on delineating the relationship between fixation points and the PLD. The mean Euclidean distances between the fixation point and the pelvic center were derived. Similar to the gaze event analysis, only data from the Complete dynamic condition were used. A mixed-design ANOVA was performed with one between-subject factor (Experience) and one within-subject factor (Recording Type). There was a significant main effect of Experience, $F(2, 34) = 4.50, p < 0.05, \eta_p^2 = 0.21$. Post-hoc pairwise comparisons showed that the Euclidean distance for NDs (mean = 165 pixels, SE = 10.35) was significantly greater than that for VDs (mean = 123 pixels, SE = 10.35, $p < 0.01$), and its difference to that for TDs approached the conventional level of significance (mean = 133.10 pixels, SE = 9.95, $p = 0.083$). There was no difference between VDs and TDs ($p = 0.76$). This finding indicates that, on average, VDs and TDs tended to focus more on the movement of the pelvic center when performing the perceptual judgment. There was also a significant main effect of Recording Type, $F(1, 34) = 11.21, p < 0.01, \eta_p^2 = 0.25$. The average fixation locations were closer to the pelvic center in the Air condition (mean = 130 pixels, SE = 5.01) than in the Ground condition (mean = 151 pixels, SE = 8.01, $p < 0.01$). This result suggests that with the more difficult air movements, the participants tended to focus more on the dancer's pelvic area. Finally, the interaction between Experience and Recording Type was not significant, $F(2, 34) = 1.92, p = 0.16, \eta_p^2 = 0.10$.

As shown in Fig 3, compared to NDs, TDs and VDs generally fixed closer to the pelvic center when viewing the animations. Combined with VDs' greater sensitivity to the inverted air movements compared to TDs and NDs, this finding indicates that VDs focused more on the overall dynamic patterns of the pelvic movement, extrapolated based on the hip joints when viewing the PLD. Furthermore, although there was no significant difference between TDs and VDs in terms of their fixation locations, due to a lack of experience with vertical dance, TDs had an intrinsic bias towards ground movements (see Fig 2). As a result, even though TDs may potentially be able to identify the relevant visual information from the display based on their comparable gaze behaviors as VDs, they still failed to correctly identify the artificial inversion, as demonstrated in the d' analysis.

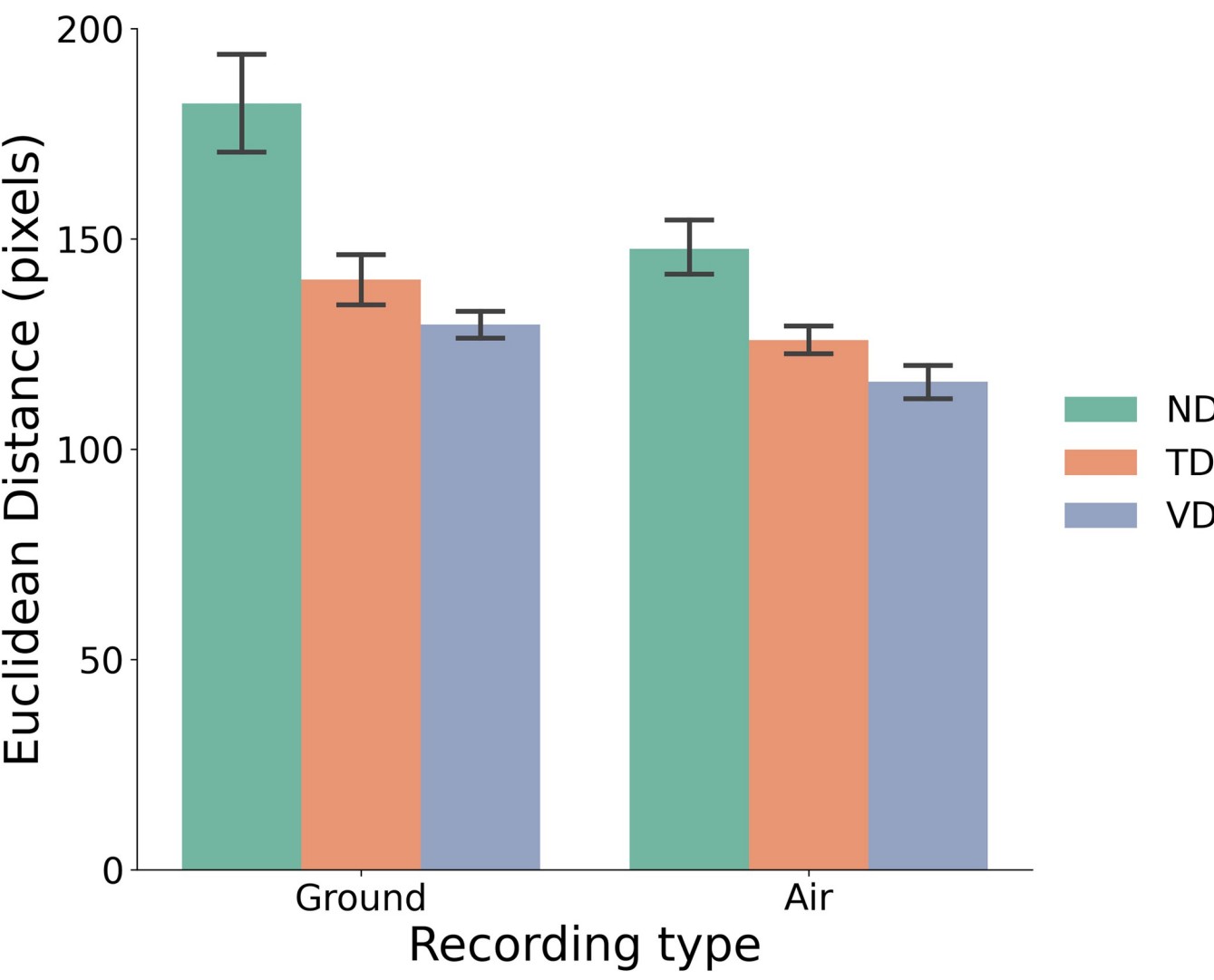

**Fig 3.** Mean Euclidean distance (in pixels) between the fixation point and the pelvic center for participants with different experience levels (hue; ND: Non-dancers, TD: Typical dancers, VD: Vertical dancers) for the ground and air movements. The pelvic center was derived as the midpoint of the two hip joints. Error bars represent the standard error of the means.

## General discussion

Pondering the inversion effect in biological motion perception, the current study examined whether the extended visuomotor experience with inverted movements could help the observer restore the human movement perception that was contaminated by the inversion. Working with professional VDs, Wang et al. [14] systematically manipulated the orientation of dance movements in PLDs and showed that only VDs, but not TDs or NDs, could overcome the inversion effect and distinguish the physically inverted (i.e., performed in the air) from artificially inverted movements when given sufficient dynamic information. Using the same experimental paradigm with a different group of participants, the current study not only replicated the previous finding where the extended visuomotor experience with inverted movements could remedy the inversion effect on perceiving human actions, but also extended the knowledge by showing that VDs engaged in longer fixations during the task to extract the dynamic information from the pelvic girdle for successful judgement. Interestingly, TDs

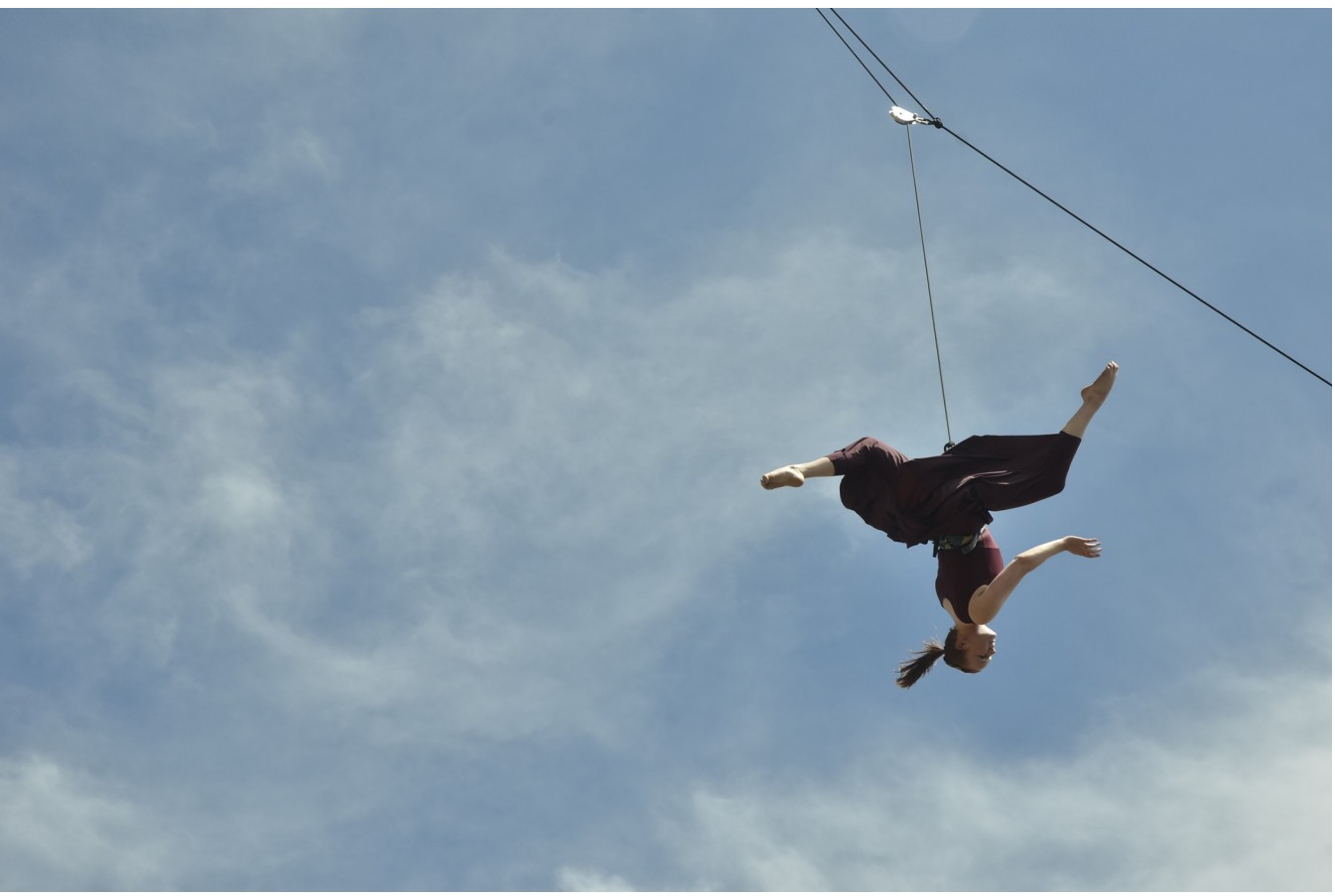

**Fig 4. An example of a dancer performing a vertical dance movement while hanging upside down in the air.**

demonstrated a similar fixation pattern as VDs, but they failed to extrapolate the underlying dynamic information in the movement to improve their sensitivity to the inversion. The comparisons between the three participant groups suggested two factors that could have contributed to VDs' ability to identify artificially inverted movements: the fixations at the pelvic girdle to extract dynamic information from the kinematics, and the prior experience that facilitates the interpretation of the attended dynamic information.

With vertical dance, the dancers wear a climbing harness placed at the waist slightly above the iliac crest. They use this harness and a rope to be suspended in the air while performing various dance movements either in an upright or an inverted orientation (Fig 4). In a normal performance, the dancers would sway from the rope, resulting in the underlying dynamics of the overall movements being that of a pendulum:

$$\frac{d^2\theta}{dt^2} + \frac{g}{l}\sin\theta = 0$$

Where $t$ is time, $\theta$ is the angle from the vertical to the pendulum, $g$ is the gravitational constant, and $l$ is the length of the line. Noticeably, only the gravitational force dictates the movement (kinematics) of the pendulum. In contrast, for dance movements performed on the ground, the dynamics of the pelvis are governed by gravity and are subject to the influence of the force generated by the dancer's limbs. The difference in the available force between the air

and ground movements in the current study was manifested around the pelvis. Since the vertical dancer is attached to the rope through a harness at the pelvis, the kinematics of pelvic motion specifies the pendulum dynamics of a particular dance movement. Combined with the more expressive movements of the limbs, this unique dynamic configuration reflects the interplay between the constraints imposed by the harness and the rope and the variability of the movements of the limbs. Conversely, when dancing on the ground, the pelvic motion would be more variable without the constraints of the harness with its kinematics specifying the dynamics of gravity and that of the dancer's self-initiated movement.

Given the air and ground movements' unique dynamic signatures, the perceptual consequences of presenting these movements in an inverted orientation could also vary. For movements performed in the air, only the direction of the embedded gravitational force in the pendulum dynamics would be inverted. Because the professional VDs who recorded the original displays were able to maximally control their limb positions when performing movements in the air, movements of their limbs in the display did not contain sufficient information on the pendulum dynamics, requiring the observers to rely mainly on the movement of the pelvis. In contrast, for movements performed on the ground, not only was the direction of gravity inverted, but, more importantly, the direction of the self-initiated force (e.g., using legs to push one off the ground) was also inverted. Therefore, the observers could rely on both movements of the torso and the limb to perform the judgment. Because of the richness of the dynamic information of the ground movements, participants of different experience levels could all readily identify the artificial inversion. Furthermore, compared to NDs, both VDs and TDs had significant dance experience, which includes education and knowledge of the role of the center of mass in postural control and dance performance [25,26]. Therefore, with a lack of experience or knowledge, NDs failed to pay attention to the pelvis to extract relevant dynamic information from the air movements. Curiously, although both VDs and TDs were able to fixate on the pelvis for the air movements, only VDs were able to successfully distinguish artificial inversion from physical inversion of the movements. What could have contributed to this discrepancy?

The mere knowledge of the information-rich area to attend does not guarantee effective information pick-up. The observations that 1) VDs and TDs had comparable eye movement patterns and 2) VDs had greater sensitivity to artificial inversion of air movements suggest that VDs were able to utilize the kinematics of the pelvis to extrapolate the embedded pendulum dynamics and leverage the dynamics to perform directional judgment, whereas TDs were not able to do so. In a recent meta-analysis, Silva and colleagues [27] found that novices and experts did not have distinct gaze patterns in the context of sports but there were still distinct performance outcomes between the two groups. In other words, the general expertise (i.e., dance) may affect gaze behavior, and the area-specific expertise (i.e., vertical dance) dictates how effectively one could pick up and interpret the information obtained from fixations [28]. While the current study did not address the question of how the dynamic information is interpreted, future studies could adopt a perturbation paradigm and systematically alter the kinematics of the pelvis of the dancer in different directions to see how these changes would affect VDs' ability to perform the judgment. Similarly, because TDs are already familiar with the area around which they should attend to obtain the relevant dynamic information, future studies could also investigate the perceptual learning process through which TDs could learn how to interpret the dynamic information to perform the judgment.

In conclusion, the findings from this study highlight the complex interplay between perceptual expertise and extracting dynamic information from the kinematics (i.e., kinematic specification of dynamics) in the context of biological motion perception, particularly focusing on the inversion effect. Judgment results showed that extended visuomotor experience with

vertical dance played a pivotal role in helping VDs overcome the inversion effect of PLDs. Shown through the eye tracking analysis, the ability of VDs to discern artificially inverted movements was underpinned by their adeptness in fixating on the pelvic area to extract dynamic information critical for judgment. Intriguingly, while both VDs and TDs exhibited similar eye movement patterns, only VDs demonstrated heightened sensitivity to artificial inversion, suggesting that area-specific expertise strongly influences information interpretation and subsequent perceptual outcomes. These insights underscore the nuanced nature of expertise-driven gaze behavior and emphasize the importance of considering domain-specific knowledge in understanding perceptual processes. Moving forward, future research could delve into the mechanisms underlying dynamic information interpretation and explore perceptual learning processes to enrich our understanding of expertise-mediated perceptual phenomena. By probing deeper into these issues, the intricacies of human perception and cognition can be unraveled and provide valuable guidelines in training and education.

## Supporting information

**S1 File.**
(DOCX)

## Acknowledgments

We would like to thank Hiroyuki Mitsudo and Kylie A Steel for their constructive feedback during the review process.

## Author Contributions

**Data curation:** Xiaoye Michael Wang, Zhichen Feng, Mingming Yang, Jing Samantha Pan, Qin Zhu.

**Formal analysis:** Xiaoye Michael Wang.

**Investigation:** Xiaoye Michael Wang, Zhichen Feng, Mingming Yang, Jing Samantha Pan, Qin Zhu.

**Methodology:** Xiaoye Michael Wang, Margaret A. Wilson, Qin Zhu.

**Project administration:** Qin Zhu.

**Software:** Xiaoye Michael Wang.

**Visualization:** Xiaoye Michael Wang.

**Writing – original draft:** Xiaoye Michael Wang.

**Writing – review & editing:** Xiaoye Michael Wang, Margaret A. Wilson, Qin Zhu.

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
