## [Decision Letter · Decision Letter 0]

12 Nov 2024

PONE-D-24-39075Experience Modulates Visual Search Strategies and the Effectiveness of Information Pickup to Overcome the Inversion Effect in Biological Motion PerceptionPLOS ONE

Dear Dr. Wang,

Thank you for submitting your manuscript to PLOS ONE. After careful consideration, we feel that it has merit but does not fully meet PLOS ONE’s publication criteria as it currently stands. Therefore, we invite you to submit a revised version of the manuscript that addresses the points raised during the review process. Your manuscript has been reviewed by two experts in the field. While their comments are favorable, there are a few concerns that need to be adequately addressed to further strengthen your paper.

We look forward to receiving your revised manuscript.

Kind regards,

Shenbing Kuang, Ph.D.

Academic Editor

PLOS ONE

Journal Requirements:

3. Please include a complete copy of PLOS’ questionnaire on inclusivity in global research in your revised manuscript. Our policy for research in this area aims to improve transparency in the reporting of research performed outside of researchers’ own country or community. The policy applies to researchers who have travelled to a different country to conduct research, research with Indigenous populations or their lands, and research on cultural artefacts. The questionnaire can also be requested at the journal’s discretion for any other submissions, even if these conditions are not met.  Please find more information on the policy and a link to download a blank copy of the questionnaire here: https://journals.plos.org/plosone/s/best-practices-in-research-reporting. Please upload a completed version of your questionnaire as Supporting Information when you resubmit your manuscript.

5. You have indicated that data is available from [njperson@uwyo.edu.].  Please can we ask you to provide us with a general contact email address for the data requests, so readers can request access in perpetuity. If a general email is not available please provide a link to a website where readers can obtain access to data.

Reviewers' comments:

Reviewer's Responses to Questions

**Comments to the Author**

1. Is the manuscript technically sound, and do the data support the conclusions?

Reviewer #1: Partly

Reviewer #2: Yes

2. Has the statistical analysis been performed appropriately and rigorously? 

Reviewer #1: I Don't Know

Reviewer #2: Yes

3. Have the authors made all data underlying the findings in their manuscript fully available?

Reviewer #1: Yes

Reviewer #2: Yes

4. Is the manuscript presented in an intelligible fashion and written in standard English?

Reviewer #1: Yes

Reviewer #2: Yes

5. Review Comments to the Author

Reviewer #1: Review of "Experience Modulates Visual Search Strategies and the Effectiveness of Information Pickup to Overcome the Inversion Effect in Biological Motion Perception" by Wang et al.

Wang et al. performed a psychophysical experiment to examine the effect of training on judging biological motion. The authors used biological motion stimuli created from vertical dancers and analyzed participants' behavioral judgements and eye movements. The vertical dancers were sensitive to the figural artificial inversion and showed a specific pattern of fixations. The experiment was creative and well-designed. However, I have several concerns as described below.

Title (and throughout the paper)

The use of "visual search (strategies)" in this paper is misleading because the visual search literature is generally involved in situations where participants look for something. In this study participants were not explicitly asked to find an item among discrete items.

p. 6, 2nd para "Ponmanadiyil and Woolhouse (2018)"

The paper did not appear in the reference list.

p. 6, 3rd para "faster saccades"

This description is a bit misleading because the velocity of saccades was not analyzed in this study.

p. 9 1st para

The phrase "The experiment was presented on ..." should probably read a phrase like "The experimental stimulus was presented on ..."

p. 11, "Each group contained 10 trials with responses of yes for the displayed movement was artificially inverted and no for the movement was not inverted."

The phrases "was artificially inverted/was not inverted" are not clear to me.

pp. 11-12, "The locations of the fixation and the locations of each point in PLD were normalized based on the screen’s resolution, resulting in values between 0 and 1."

Readers will be also helpful to know how long the actual distance is. On p. 9, the authors say "... on a Dell 22-inch monitor (E2222H) with a 1680 × 1050 resolution." Then, does a normalized distance of 1 (maximal value) correspond to actual 22 inches?

p. 14, the 4th line from the bottom

"latter" should probably read "former."

p. 15, Figure 2

I am confused about that the baseline of bar graphs is d' = -1.0, not 0.0. d' = 0.0 is the reasonable baseline of sensitivity as the authors indicate on p. 11. Furthermore, on reading the vertical axis of the Congruent air condition, I found that mean VD = -0.7, TD & ND = ~-1.0. These values are unlikely to be consistent with the main text on p. 14, as the authors wrote "When the movement was performed in the air, VD (mean = 0.63, SE = 0.16) showed greater sensitivity than TD (mean = 0.04, SE = 0.16, p < 0.05) and ND (mean = 0.08, SE = 0.16, p = 0.06), ...." Am I missing something?

p. 18, "Spatial Analysis"

The authors performed an ANOVA with a factor of Joint on the normalized distances. I am concerned about the different joint conditions of normalized distances can be regarded as statistically independent from each other (i.e., as one of assumptions for performing an ANOVA) because the values for different joints were calculated based on the same gaze point at each time. For example, when normalized distances for the torso center becomes shorter, that for the head will be mathematically longer at that time since the same fixated gaze position was assumed. This kind of mathematical dependence might inflate the chance of rejecting the statistical null hypothesis. The authors may verify that performing ANOVAs on such a situation is valid or tolerable by citing appropriate statistical methodology articles or similar papers.

p. 16, line 8

"former" should probably read "latter."

p. 20, first line

Typo: "Error! Reference source not found"

p. 21, Figure 3

"(x-axis)" is a bit misleading because it implies the coordinates of the screen for the PLD presentation.

Reviewer #2: This paper, which explores biomotion in the context of traditional versus vertical dance, is exceptionally well-written and clear. The findings advance our understanding of biological motion and the perceptual factors that inform it. I have provided only minor comments throughout, which should be straightforward to address. Congratulations on presenting a thoughtfully constructed and insightful study.

6. PLOS authors have the option to publish the peer review history of their article (what does this mean?). If published, this will include your full peer review and any attached files.

Reviewer #1: **Yes: **Hiroyuki Mitsudo

Reviewer #2: **Yes: **Kylie A Steel

---

## [Author Response · Author response to Decision Letter 0]

9 Dec 2024

We attached response to reviewer comments as a reparate document.

---

## [Decision Letter · Decision Letter 1]

26 Dec 2024

Experience Modulates Gaze Behavior and the Effectiveness of Information Pickup to Overcome the Inversion Effect in Biological Motion Perception

PONE-D-24-39075R1

Dear Dr. Wang,

We’re pleased to inform you that your manuscript has been judged scientifically suitable for publication and will be formally accepted for publication once it meets all outstanding technical requirements.

Kind regards,

Shenbing Kuang, Ph.D.

Academic Editor

PLOS ONE

Additional Editor Comments (optional):

Reviewers' comments:

Reviewer's Responses to Questions

**Comments to the Author**

1. If the authors have adequately addressed your comments raised in a previous round of review and you feel that this manuscript is now acceptable for publication, you may indicate that here to bypass the “Comments to the Author” section, enter your conflict of interest statement in the “Confidential to Editor” section, and submit your "Accept" recommendation.

Reviewer #1: All comments have been addressed

Reviewer #2: All comments have been addressed

2. Is the manuscript technically sound, and do the data support the conclusions?

Reviewer #1: Yes

Reviewer #2: Yes

3. Has the statistical analysis been performed appropriately and rigorously? 

Reviewer #1: Yes

Reviewer #2: Yes

4. Have the authors made all data underlying the findings in their manuscript fully available?

Reviewer #1: (No Response)

Reviewer #2: Yes

5. Is the manuscript presented in an intelligible fashion and written in standard English?

Reviewer #1: (No Response)

Reviewer #2: Yes

6. Review Comments to the Author

Reviewer #1: Second review of "Experience Modulates Gaze Behavior and the Effectiveness of Information Pickup to Overcome the Inversion Effect in Biological Motion Perception" by Wang et al.

The authors adequately and sufficiently responded to almost all comments made by myself. Only one comment on the Perceptual Judgment section:

p. 14, “When the movement was performed in the air, VD (mean = 0.63, SE = 0.16) showed greater sensitivity than TD (mean = 0.04, SE = 0.16, p < 0.05) and ND (mean = 0.08, SE = 0.16, p = 0.06), albeit the latter only approached the traditional level of significance.”

p. 16, “However, for the Complete condition, VD (mean = 1.32, SE = 0.15) had greater sensitivity than TD (mean = 0.85, SE = 0.14, p = 0.073) and ND (mean = 0.43, SE = 0.15, p < 0.001), albeit the former only approached the traditional level of significance.”

I now find that my problem in interpreting these sentences is the phrase "only approached." These sentences imply that the other comparison "did not approach" (but clearly reached) the traditional level of significance. On p. 14, for example, it might be better to write like "..., albeit the latter was slightly above the traditional level of significance."

Reviewer #2: I have reviewed the authors responses to both reviewers and am satisfied with the ammendments that the authors have made.

7. PLOS authors have the option to publish the peer review history of their article (what does this mean?). If published, this will include your full peer review and any attached files.

Reviewer #1: **Yes: **Hiroyuki Mitsudo

Reviewer #2: **Yes: **Kylie A Steel

---

## [Editor Report · Acceptance letter]

17 Jan 2025

PONE-D-24-39075R1 

PLOS ONE

Dear Dr. Wang, 

I'm pleased to inform you that your manuscript has been deemed suitable for publication in PLOS ONE. Congratulations! Your manuscript is now being handed over to our production team.

Kind regards, 

on behalf of

Associate Professor Shenbing Kuang 

Academic Editor

PLOS ONE